# Fabric Impregnation with Shear Thickening Fluid for Ballistic Armor Polymer Composites: An Updated Overview

**DOI:** 10.3390/polym14204357

**Published:** 2022-10-16

**Authors:** Matheus Pereira Ribeiro, Pedro Henrique Poubel Mendonça da Silveira, Fábio de Oliveira Braga, Sergio Neves Monteiro

**Affiliations:** 1Department of Materials Science, Military Institute of Engineering—IME, Praça General Tibúrcio 80, Urca, Rio de Janeiro 22290-270, Brazil; 2Department of Civil Engineering, Federal Fluminense University—UFF, Niterói 24210-240, Brazil

**Keywords:** shear thickening fluid, ballistic fabric, ballistic performance, energy absorption, ballistic limit

## Abstract

As destructive power of firearms raises over the years, ballistic armors are in continuous need of enhancement. For soft armors, this improvement is invariably related to the increase of stacked layers of high-strength fiber fabrics, which potentially restrains wearer mobility. A different solution was created in the early 2000s, when a research work proposed a new treatment of the ballistic panels with non-Newtonian colloidal shear thickening fluid (STF), in view of weight decreasing with strength reinforcement and cost-effective production. Since then, databases reveal a surge in publications generally pointing to acceptable features under ballistic impact by exploring different conditions of the materials adopted. As a result, several works have not been covered in recent reviews for a wider discussion of their methodologies and results, which could be a barrier to a deeper understanding of the behavior of STF-impregnated fabrics. Therefore, the present work aims to overview the unexplored state-of-art on the effectiveness of STF addition to high-strength fabrics for ballistic applications to compile achievements regarding the ballistic strength of this novel material through different parameters. From the screened papers, SiO${}_{2}$, Polyethylene glycol (PEG) 200 and 400, and Aramid are extensively being incorporated into the STF/Fabric composites. Besides, parameters such as initial and residual velocity, energy absorbed, ballistic limit, and back face signature are common metrics for a comprehensive analysis of the ballistic performance of the material. The overview also points to a promising application of natural fiber fabrics and auxetic fabrics with STF fluids, as well as the demand for the adoption of new materials and more homogeneous ballistic test parameters. Finally, the work emphasizes that the ballistic application for STF-impregnated fabric based on NIJ standards is feasible for several conditions.

## 1. Introduction

Since earlier civilizations, humans have been trying to protect themselves from all possible types of menace, such as natural disasters and predators. The scarcity of important resources such as food, land, and other utilities has always been a source of war and armed conflicts. As a consequence, this phenomenon induced the development of weapons, such as swords, axes, firearms, as well as protective armor [1,2]. Over the centuries, several materials have been employed to protect the human body from threats, e.g., wood, leather, wool, bronze, and steel [3,4]. As the destructive power of threats raises, modernly related to firearms, attention to research in the field of advanced materials has been drawn, due to the demand of combining a high-level protection and mobility [5]. Related to materials technologies, important differences emerge as advancements are made over time. For example, materials for body armor changed drastically during the two world wars of the twentieth century: During World War I, the combatants wore metal helmets and body plates. In World War II, nylon was implemented in several military apparatus, as well as fiberglass, and polyaramid, owing to a relatively lower density and higher strength [5,6], allowing mobility to the combatants.

From 1960s on, studies on light ceramic armor and polymeric materials started to gather relevance in the use of body armor plates made up with alumina (Al_2_O_3_), silicon carbide (SiC) and boron carbide (B_4_C), prevailing in hard armors systems [7,8,9,10], due to high impact strength [11,12]. At the same time, soft armors emerged based on high-performance polymeric fabrics and polymer matrix composites (PMC) in a ballistic application for lighter gears. In terms of classification, different categories are listed according to the National Institute of Justice (NIJ) [13] in Table 1. Soft armor is classified in the lower and medium levels in the NIJ standard, IIA, II, and IIIA, in which protection requirements are against low-energy projectiles, such as the bullets shot from handguns [14,15]. The higher levels must include hard armor inserts in what is called “in conjunction design”. Those plates are destined for high-risk military applications, since they are subjected to higher energy projectiles, such as the bullets from AK-47s.

Soft armor is ballistic personal protection made of several stacked layers of high-strength woven fiber fabrics, either sewn or glued together using low-modulus binders, such as elastomers or soft plastics. The main objective of this piece of armor is to provide ballistic protection to the user without restraining its mobility. To achieve its goal, high-performance synthetic fibers fabrics are used, such as carbon fibers, aramid, fiberglass, and nylon [16,17,18,19,20,21,22].

Since the early development of Kevlar^®^ by DuPont in the mid-1960s, investigation of the ballistic performance of soft armor materials has been increasing exponentially. New high-strength fibers have gradually replaced nylon as fabrics for military applications. From the so-investigated aramid fiber, other types of fabrics, besides the well-known Kevlar, have been developed, such as Twaron and Technora^®^ fabrics [23,24]. Besides, ultra-high molecular weight polyethylene (UHMWPE) fabrics also presents interesting ballistic performance, and is already being commercialized by several manufacturers as products such as the Spectra and Dyneema^®^ fabrics [25,26,27].

As a limitation, several works report that to achieve acceptable ballistic strength in soft armors, the number of required fabric layers in a typical panel can be as high as 40. This represents a considerable increase in the armor weight, thickness, and stiffness of the final gear. In other words, the mobility of the user is still an issue for engineers and researchers [28,29,30,31]. Regarding this problem of the excessive number of layers, an interesting solution was created in the early 2000s. A research group from the University of Delaware and U.S. Army Laboratory developed an aramid panel with fabric layers treated with a non-Newtonian colloidal shear thickening fluid (STF) [32,33]. The idea of using STF in ballistic applications originated from a technical report entitled “Evaluation and development of fluid armor system” by Gates Jr. (1968) [34]. The author had the objective to develop a fully-flexible material that would be able to self-heal after impact. An important finding was that powdered silica mixed with a liquid carrier improved the fragment penetration strength into the layered flexible fabrics. Nonetheless, the first investigation reporting the use of STFs as aramid fabrics filling was only published in 2003 by Lee et al. [35], who also submitted a patent in 2003 [32]. The authors reported that the impregnation of Kevlar with STF decreased the number of necessary layers from 10 to 4, necessary to reach the same ballistic performance previously reached.

Figure 1 presents the number of publications in recent years in the field of STF for ballistic applications, represented by the red line, compared to simply STF as a subject since the late 1960s, represented by the black line. The interest of researchers in the former topic has been growing exponentially manner until today, while the latter grows linearly until today.

There are two antagonistic ways to understand this information: (1) STFs do not improve the properties of fabrics in ballistic applications, and thus the number of publications will drop in the following years; (2) There is an increasing lack of scientific research on the subject “STF for ballistic applications”. The authors of the present work find the second alternative more plausible, which is confirmed by the promising results found by several researchers in the field of STFs for ballistic applications in recent years [36,37,38,39,40,41]. Thus, deeper and wider investigations are needed to explore their full potential, as these papers represent a very limited fraction of all STF related works. Attempts to review the subject have been made until recently. Gürgen, Kuşhan and Li [38], Mawkhlieng and Majumdar [2], Khodadadi et al. [42], as well as Zhang et al. [43] are good examples. However, their extensive reviews do not cover the entire literature in the field, so it is essential to perform a review that fills in these gaps and covers important papers that have been left out. Therefore, the objective of the present work is to review the state-of-art on STFs as an improvement to high-strength fiber fabrics for ballistic applications in a way to contribute to the advancement of the field by compiling and discussing relevant topics and parameters related to materials, methodologies, and results achieved by unexplored reports. For this, the theoretical background is given and a future trend is analyzed based on recent progress.

## 2. Theoretical Basis

### 2.1. Shear Thickening Behavior

Shear thickening fluids (STF) are a type of non-Newtonian fluid whose viscosity increases drastically as the shear rate or shear stress increases [44]. The thickening behavior of these fluids is observed in colloidal suspensions composed of solid particles, called dispersed phase, e.g., silica, calcium carbonate, and boron carbide, to mention a few, which are inserted in carrier fluids, called dispersion phase, e.g., water, ethylene glycol, polyethylene glycol. The dispersion phase presents, in general, a linear Newtonian behavior, so the mixing with the solid particles will allow the shear thickening mechanism [45,46,47]. Figure 2 illustrates the rheological behavior of a typical STF.

In Zone I, the particles are randomly dispersed in the medium at equilibrium. As the shear rate increases, the particles tend to organize as layered structures, which results in shear thinning, the opposite of shear thickening, with a consequent reduction in viscosity, indicated as Zone II in Figure. However, along with the increasing shear rate, the apparent viscosity reaches a threshold characterized by Zone III, indicating the shear thickening onset. The mechanism behind this zone can be explained by the disordering in those layered structures, which tend to form agglomerated particles resulting from hydrodynamic lubrication forces, called hydroclusters, responsible for this radical viscosity change [48].

According to previous review works, several research papers proposed different theories to explain the shear thickening behavior of these fluids. Table 2 summarizes the most acceptable explanations.

The benefit of shear thickening behavior of fluids in the mechanical and ballistic applications was reported in numerous studies [51,52,53,54,55,56,57,58,59,60,61,62,63,64,65]. For these studies, a thickening ratio is taken into account in the efficiency of an STF [46]. Different parameters related to the phases of the STF, dispersion, and dispersed lead to a variation in viscosity. Table 3 summarizes the different parameters that influence the apparent viscosity of the material.

### 2.2. Ballistic Fabrics Impregnated with STF

Two-dimensional (2D) fabrics are commonly the most used and explored body protection in ballistic applications. They are composed of fixed yarns called warp, while the weft yarns are inserted under and over them [28,43,79,80]. Typically, as mentioned before, aramid, UHMWPE, polyphenylenebenzobisoxazole, and nylon are used in soft armor to provide protection from bullets [36,81,82,83,84] and stabbing [52,54,55,85,86,87], as they present high-strength/modulus fibers and offers the proper strength to impact and trauma according to current security norms. Table 4 presents the parameters related to fabric that influence the ballistic performance.

Karahan et al. [92] compared the ballistic energy absorption for the two most used types of materials: aramid and UHMWPE fabrics, set with 24 layers each. Numerically, the energy absorption of all aramid fabrics was greater than all UHMWPE fabrics. However, when weight is taken into account through the energy absorption and areal density ratio, all UHMWPE fabrics presented greater values than aramid. Othman and Hassan [93] investigated the ply number variation. They presented results of ballistic tests in which the energy absorption of neat aramid fabric increased from 1.48% to 78.04% just varying from a single layer to a 10-layer system, reaching 100% with 15 layers. Furthermore, Tan et al. [94] reported an improvement in the ballistic limit of Twaron impregnated with silica colloidal water suspension with the increase in ply number from one to four, considering the same impregnation volume fraction. Considering ply orientation, Arora et al. [26] concluded that angular-orientated fabrics, with 12.5° and 15° of pace, performed better under ballistic impact than aligned fabrics, reaching up to 58% enhancement in energy absorption. Different research papers [95,96,97,98] found interesting results with chemical and physical treatment for the improvement of fibers’ chemical structures of ballistic fabrics. Chu et al. [97] performed a yarn-pullout test, closely related to the ballistic property, in Twaron fabrics treated with plasma-enhanced vapor deposition with dichlorodimethylsilane ((CH${}_{3}$)${}_{2}$Cl${}_{2}$Si) under different time exposures. They found that the coefficient of static friction increased by about 84% for 4 min of treatment. Sun et al. [98] reached about 300% of strength to pulling out yarns from Kevlar fabric also treated with (CH${}_{3}$)${}_{2}$Cl${}_{2}$Si but for 5 min.

Although the aforementioned treatments provided enhancement to ballistic fabric, a different approach was envisioned in 2003 by Lee et al. [35], when they proposed the use of STF to increase the ballistic strength of ballistic fabric without sacrificing areal density and flexibility. Their composite was made by the immersion of the fabric in the viscous solution, which acts between the yarns, increasing friction and shear thickening reaction to shear stress. The preparation scheme of the STF-impregnated composites is illustrated in Figure 3. Initially, the particles are mixed into a dispersing phase, resulting in a homogeneous suspension and disaggregated particles, with a mechanical, magnetic, or ultrasonic device [46,99]. Other devices have also already been used, such as an airbrush [100]. The viscous fluid is diluted in alcohol in a fixed proportion and mixed again to convert it into an easily dispersive liquid for uniform impregnation into the interstices of the fabric. After that, the fabric is finally soaked into the solution for a preset time and dried until the alcohol is volatilized. Different works suggest a padding process before drying for removing excess fluid, indicating an increase in yarn friction [101,102] due to a decrease in STF add-on, resulting in a more balanced friction factor, preventing the fabric from an easy pullout or fiber break under impact.

While reviewing textile-based soft armors in general, Mawkhlieng and Majumdar [2] affirmed that the main energy absorption mechanisms in a ballistic fabric are out-of-plane deformation, yarn pullout, yarn rupture, fibrillation, friction and bowing. Crouch [103] highlights the first two mechanisms, relating them to the cross and pyramid aspects, respectively, as presented the Figure 4. Khodadadi et al. [42] stated that a combined mechanism of primary yarn tension, fabric deformation, frictional yarn/projectile slipping, yarn/fabric pullout, and fiber breakage dissipates the impact kinetic energy in fabric armors. For a ballistic fabric impregnated with STF, Srivastava et al. [70] emphasize three energy absorption mechanisms related to the liquid–fabric interaction: inter-yarn friction, load transfer, and shear thickening mechanism. Gürgen and Kushan [46] affirm that the influence of the shear thickening mechanism of non-newtonian fluids is not clear in several works. The authors attribute the mechanical strength enhancement primarily to friction along woven fabrics, which causes the load transfer to spread over a wider area and lower penetration depth. Despite this assertion, another work by Mawkhlieng and Majumdar [104] analyzed different STFs that provided similar yarn pullout strength in an aramid fabric. The impregnation of fabric with the fluid with a higher thickening ratio (maximum viscosity to viscosity at the critical shear rate) resulted in higher impact strength, definitely evidencing the influence and contribution of the shear thickening mechanism on ballistic performance.

For a more suitable observation in real-life applications, the ballistic tests cover this requirement well [104]. This kind of analysis for soft armor is generally prepared based on the NIJ standards [106], which classify the material in protection levels, as presented above in Table 1, based on the projectile caliber, geometry, material types, mass, velocity, and allowable back face signature (BFS) [79]. Karahan et al. [92] detailed an actual high-velocity impact ballistic test, which adopts a few equipments, such as a gun barrel, to fire the projectile. Additionally, it used chronographs to measure the velocity of the projectile at different distances, clamping material, and plastilina clay to support the target when the study of BFS is required. Figure 5 presents a scheme of a ballistic test set.

According to Morye et al. [107], from the measured velocity before and after impact, in the case of projectile perforation, the energy absorbed by the fabric, an important parameter for ballistic performance, can be analyzed through the law of conservation of energy by the following Equation (Equation 1).
(1)$$E_{abs}\left(J\right)=\frac{1}{2}\cdot m\cdot \left(v_{i}^{2}-v_{r}^{2}\right)$$
where *m* is the mass of the projectile, $v_{i}$ is the initial velocity, before impact, and $v_{r}$ is the residual velocity after impact.

Another assessment aspect of the ballistic performance of the soft armors widely observed by research papers is the BFS or blunt trauma [103]. It is a vital remark for materials that keep the integrity after an impact to some degree. A behind armor blunt trauma (BABT) can also lead to a fatal injury for the user [108]. This parameter is measured from the clay witness placed right behind the target. After impact, the material undergoes a trauma with a specific volume, depth, or diameter to dissipate the energy of the projectile and stop its penetration [42]. For this purpose, NIJ standards suggest a 44 mm (1.73 inch) BFS.

For the STF-impregnated fabrics, the specific energy absorption (SEA) and the energy enhancement ratio (EER) appear as interesting parameters for weight/energy absorption equilibrium, as the final load of the equipment is directly related to the user performance. The SEA parameter expresses the ballistic performance enhancement related to the weight of the gear. The EER values also indicate how much energy the material could absorb without increasing its weight, but it is applied in comparison among groups of samples. Both parameters are reported in Mishra et al. [81], but Liu et al. [109] refer to the same parameter as “ballistic performance index” (BPI). The SEA is given by Equation (Equation 2):(2)$$SEA\left(Jm^{2}/kg\right)=\frac{E_{abs}}{\rho_{A}}$$
where $\rho_{A}$ is the areal density. The EER parameter, also reported by Mishra et al. [81] and Liu et al. [109], is defined in Equation (Equation 3):(3)$$EER=\frac{SEA_{f}}{SEA_{i}}$$
where $SEA_{i}$ is the SEA of the neat fabric or any reference group, and $SEA_{f}$ is the SEA of the impregnated fabric. Despite these parameters, it is worth mentioning that the ballistic response depends on several other features, such as those reported by Mawkhlieng and Majumdar [2]:Impact velocity;Distance between target and gun barrel;Boundary conditions;Angle of incidence;Point of impact;Shot location.

## 3. State-of-the-Art

### 3.1. Research Works Selection

In light of the foregoing, the present work focused essentially on recently published and previously unexplored papers from 2003 to 2022, since the first report exploring the ballistic performance of an STF-impregnated fabric composite is dated 2003 [35]. Table 5 lists all the review works screened and analyzed for the selection of unexplored reports, as well as the most relevant works, cited and reviewed related to the ballistic analysis of fabrics impregnated with STF.

The research collection was updated until September 2022 through the electronic databases: SCOPUS, Web of Science, Google Scholar, SciELO. Initially, the investigation was performed based on a combination of terms and Boolean operators, limited to research and original articles: ((“shear thickening fluid”) OR (“stf”)) AND (“ballistic”). After a thorough screening of all the works found, however, only a limited number of papers did present the ballistic test in any kind of fabric impregnated or combined with STF. Thus, for a more accurate analysis, eligibility criteria were set, such as (*i*) works related to fabric impregnation with STF; (*ii*) residual velocity, BFS or energy absorption results; (*iii*) papers not cited yet in previous reviews related to STF-impregnated fabric. Figure 6 illustrates the selection process and the number of works obtained.

### 3.2. Discussion

The works found following the aforementioned criteria are listed in Table 6 by date and discussed below.

It is interesting to note that the studies presented different approaches to the use of STF. In Table 6, it is worth noticing that a great part of the papers analyzed the use of aramid fabrics (Kevlar, Twaron) impregnated with STF of SiO${}_{2}$ nanoparticles in PEG fluid. The weight fraction of the silica particles varied between 0 to 70 wt% with a greater focus on the fractions between 20 and 50 wt%. The most adopted type of PEG fluid was the one with 200 g/mol, although higher molecular weight generally promotes a higher viscosity, as already reported in previous works [183] and tested in different papers [184,185]. The works of Yeh et al. [172], Yeh et al. [171], Chang et al. [173], Zhang et al. [176], Zhihao et al. [181] and Katiyar et al. [175] carry out what was stated.

Yeh et al. [171] investigated the stabbing and ballistic behavior of Kevlar fabrics impregnated with SiO_2_/PEG STF, using different concentrations between 20 to 50 wt%. The authors conducted ballistic tests using carbon steel spheres (2 g mass, 8 mm diameter) with an initial velocity of about 180 m·s^−1^. A comparison has been made between fabrics without impregnation (4, 8, and 10 layers) and four impregnated laminates with three different volumes of STF (2, 4, and 8 mL). The ballistic test results indicated that all the four-layer-impregnated Kevlar increased the energy absorption, up to about 16% from the four-layer neat Kevlar to four-layer Kevlar with 4 ml of STF. The sample with 10 layers was the closest among the neat-Kevlar groups but still with about 6% lower energy absorption. A decrease was also observed with the addition of 8 ml of STF, in comparison with the fabric with 4 ml of STF. On the other hand, the specific energy absorption was higher for the 2-mL-STF/four-layer Kevlar between the impregnated groups, about 25% greater than the 4-mL-STF/four-layer Kevlar with only 0.5% lower energy absorption, indicating a relatively more interesting use as soft armor.

After the first work investigating the STF content in Kevlar fabrics, Yeh et al. [172] analyzed the influence of the impregnation method on the ballistic properties of impregnated Kevlar fabrics using the 40 wt% concentration, as adopted in their previous study [171]. The authors performed the impregnation by two methods. The first one impregnates STF onto the fabric by immersion and evaporation of ethanol under 80 °C. The second uses what is known as the padding method, passing the fabric through rollers impregnated with the STF. The fabric impregnated by the padding method using four layers of Kevlar fabric presented higher ballistic performance associated with the lowest penetration depth, 55%, which was lower than that obtained for the four-layer neat Kevlar, as well as higher energy absorption, 16% higher than that obtained for the four-layer neat Kevlar.

Chang et al. [173] investigated the ballistic performance of soft armors consisting of STF-impregnated Kevlar laminates. In order to produce lighter structures that meet the NIJ IIIA classification [13], in Table 1, the authors developed hybrid composites composed of layers of Kevlar fabric impregnated with polyurea, with interleaved layers of honeycomb paper impregnated with 40 wt% SiO_2_/PEG 200 STF. The ballistic test results performed with 9 mm ammunition showed that the impregnation of polyurea on Kevlar fabric resulted in a lighter material with superior ballistic performance than the panel made of neat Kevlar fabric based on the BFS. The addition of 1 layer of honeycomb paper/STF between the Kevlar layers resulted in an improvement in the ballistic strength. Compared to a panel composed of 47 Kevlar/polyurea layers, the Kevlar/honeycomb paper/STF panel showed similar ballistic performance using only 33 layers (32 Kevlar layers + 1 STF structure layer), resulting in a panel with lower thickness and density.

Zhang et al. [176] investigated the ballistic performance of single-layer Kevlar fabrics impregnated with SiO_2_/PEG 200 STF at higher concentrations. Two concentrations were investigated: 0 wt% (neat Kevlar) and impregnated with 70 wt% STF (70 wt% STF/Kevlar). Ballistic tests were performed using 8 mm diameter spherical projectiles with an average incident velocity of 144 m·s^−1^. An interesting analysis of this paper is the fabric aspect during the impact of the projectile. Figure 7a illustrates the moment described on the neat Kevlar, recorded with a high-speed camera. It is observed that the yarn deformation is extended from the impact region of the fabric to its edge, spreading out in the transverse direction. This entire event occurs in a failure time of less than 100 μs. Figure 7b presents the relationship between the residual velocity and different incident velocities for neat Kevlar and 70 wt% STF/Kevlar. It can be seen that the impregnated fabric showed a higher ballistic limit, about 10%. The relationship of the energy absorption of the two groups of samples is displayed in Figure 7c, in which it is shown that the energy dissipation of the 70 wt% STF/Kevlar sample is higher than that of the neat Kevlar. Thus, the authors indicated that the impregnation of SiO_2_/PEG STF into Kevlar at higher concentrations such as 70 wt% is advantageous from the point of view of ballistic performance for speeds below 200 m·s^−1^.

The research conducted by Zhihao et al. [181] evaluated the influence of the temperature range between −50 and 100 °C on the impact strength of STF-impregnated Kevlar fabric, given the application as soft-wall casing of an aero-engine. They analyzed Kevlar plain-weave fabric (241.2 g/m²) and an STF composed of nanoparticles of silica (100 nm) in a 20 wt% fraction in PEG 200. They used a rectangular projectile to simulate a fractured aero-engine fan blade. The results presented a dramatic influence of temperature on both STF and Kevlar fabric, as the highest- and lowest-temperature conditions deteriorated the viscosity of the fluid and the mechanical properties of the fabric, respectively. From the ballistic tests, the authors concluded that the ballistic limit decreased to about 62.6% at −50 °C and 84.2% at 100 °C of the highest value obtained, about 67.7 m·s^−1^ under 20 °C. The energy absorption analysis presented a slight change: the maximum average energy absorption occurred at 60 °C, 400.25 J, followed by the 20 °C condition, 263.5 J; the worst average energy absorption occurred under −30 °C, 1883.43 J.

Although Katiyar et al. [175] followed the research conducted by Zhihao et al. [181], they focused on investigating the effect of functionalization of SiO_2_ particles of a SiO_2_/PEG 400 STF using aminosilane. To define the content of STF to be impregnated into the Kevlar fabric, the authors performed rheological tests to verify the thickening of the fluids as a function of increasing shear rate. The rheological tests showed that the functionalization of SiO_2_ with aminosilane resulted in a considerable increase in viscosity, along with the weight fraction of particles in STF. An unfunctionalized STF with 25 wt% SiO_2_ reached a viscosity of approximately 18 Pa.s at a shear rate of 2000 1/s, while an aminosilane functionalized STF using the same SiO_2_ content showed a viscosity of approximately 1850 Pa.s at a shear rate of less than 100 1/s. Once the functionalized STF was chosen, the ballistic performance of impregnated Kevlar fabrics was evaluated, using four and eight layers, and employing different impact velocities with a projectile of 9.5 mm diameter and 7 g mass. The ballistic results of the impregnated Kevlar panels showed that the STF contributed to an improvement in ballistic performance when compared to the panels without the presence of STF. Under an initial velocity around 154 m·s^−1^, the residual velocity decreased by about 22% with an increment of four layers, although this modification increased energy absorption by 32% [175].

The use of chemical treatment in soft armors was also investigated by Grineviciute et al. [167]. Differently from the work of Katiyar et al. [175], in which the authors chemically treated the silica particles, Grineviciute and collaborators [167] analyzed the influence of plasma treatment on Twaron fabrics and then impregnated them with different types of STF for ballistic protection in soft armors. For the impregnation, two types of STF were produced. The first was composed of orthosilicic acid (H_4_SiO_4_), and the second was composed of acrylic, both dispersed in water at proportions of 5 wt%. The ballistic tests were conducted according to NIJ standard [13], in which they used 9 mm ammunition and impact velocities of 436 ± 9 m·s^−1^. The results exhibited by the authors indicated that the employment of H_4_SiO_4_ in STF resulted in better ballistic performance, revealed by the depth of penetration measured after the ballistic test, while impregnation with acrylic reduced the ballistic absorption capacity of the fabric when compared to pure Twaron and Twaron impregnated with H_4_SO_4_. Plasma treatment on the STF by H_4_SiO_4_ potentiates the ballistic performance of the fabric. However, it causes an inferior performance of acrylic impregnation in the fabric. It is worth mentioning that the plasma treatment with STF-impregnation was also recently analyzed by Liu et al. [186] in a UHMWPE fabric subjected to a low-velocity drop-weight test, proving an enhancement of mechanical behavior.

It is also worth noting that Grineviciute et al. [167] adopt water as one of the phases for STF impregnation. Different works have already explored water as a dispersion phase in viscous fluid [23,94,101,187,188,189,190]. For ballistic application, Bablu and Manimala [177] investigated the STF-impregnated Kevlar fabrics using water as the liquid medium for SiO${}_{2}$ nanoparticle dispersion. STFs were produced with SiO${}_{2}$ concentrations of 10, 30, and 40 wt%, in which the ballistic performance was compared with neat Kevlar, varying the number of Kevlar layers used. Their ballistic tests were conducted using ammunition 0.357-inch caliber ammunition. The authors reported that the addition of 40 wt% of STF in four layers of Kevlar promoted a higher energy absorption than neat Kevlar with seven layers, resulting in a 17.3% lighter material. The use of 50 and 60 wt% concentrations in four layers also gave an interesting result, however, associated with lower weight decrease, 14.3 and 8.6%, respectively. The same occurred for fabric impregnated with the lower STF content of 10 wt% using six layers, which exhibited ballistic performance similar to neat Kevlar with seven layers with a weight reduction of only 4.7%. By contrast, fabric impregnated with 30 wt% STF exhibited superior performance using five layers but with a weight reduction of only 5.8%.

Another different liquid medium was used by Levinsky et al. [166]. The authors investigated the ballistic performance of polycarbonate (PC) plates with thicknesses of 4 and 5 mm, coupled to Twaron^®^ fabrics impregnated with STF composed of alumina (Al_2_O_3_) particles dispersed in polyvinyl acetate (PVAc). After the confection of the materials, ballistic tests were performed using 8 mm diameter steel projectiles with different initial velocities. The authors reported that the addition of one layer of Twaron+STF reduced the penetration depth by 35.5% during the test with 4 mm thick plates. For the 5 mm thick plates, the addition of the STF-impregnated fabric reduced the penetration depth by 20.8%.

Tria et al. [169], in turn, used the largely adopted PEG but did not specify the molecular weight. Their objective was to investigate the effect of the number of Kevlar layers on ballistic panels impregnated with SiO_2_/PEG STF. An unspecified concentration of SiO_2_ was used with PEG. The STF was impregnated into the Kevlar layers, using ballistic panels with 14, 16, 18, and 26 layers. Again, it was not specified how many layers were soaked into the STF solution. The ballistic tests with 9 mm ammunition and an initial velocity of 390 m·s^−1^ disclosed that the increase in the number of impregnated fabric layers resulted in a small reduction in the penetration depth after firing. On the other hand, the trauma diameter caused by the shock wave was amplified with the increase in the number of layers.

From the perspective of the panels, Mishra et al. [81], Tang et al. [18], Zhang et al. [148], Olszewska et al. [168], Bajya et al. [182], and Hasan-Nezhad et al. [178] analyzed the ballistic performance of a lightweight armor vest under other condition, namely using UHMWPE or E-Glass fabrics. Further analysis can also be observed in Nayak et al. [191], such as thermogravimetric analysis, rheological tests, and high-strain tests, which are beyond the scope of the present work.

Mishra et al. [81] evaluated the ballistic performance of UHMWPE fabrics impregnated with 40 wt% SiO_2_/PEG 400 STF panels varying the number of layers as 5, 10, 20 and 30 layers. The ballistic tests were conducted using an Al projectile, with a mass of 4g and length of 20 mm, with a velocity ranging from 250 to 700 m·s^−1^. In Figure 8, it is observed that, in comparison with the neat fabric, the STF-impregnated UHMWPE fabrics presented higher energy absorption in all conditions of projectile velocity, as well as higher values with the increase in the number of layers and impact velocity. This is attributed by the authors to the increase in the friction between fabric layers, which raises the strength against the projectile during the perforation process, in addition to the increased interaction time during the course of penetration between layers. This was the case for the samples with 20 and 30 layers of neat and STF-impregnated fabric under impact velocities greater than 500 m·s^−1^. The opposite effect occurred to the thinnest sample of five layers, which was ineffective under impact velocity higher than 450 m·s^−1^.

Mishra et al. [81] also provided an analysis based on the SEA (Equation (Equation 2)). The authors compared the respective neat fabric to the impregnated version under 509 m·s^−1^ impact and indicated that the samples with 20 layers presented the highest increment in SEA values, 10.7%, despite the small difference to the 30-layer samples, with an increment of 8.5%. It is worth noticing that they were able to produce a softer armor system when compared to Liu et al. [192].

Tang et al. [18] investigated the ballistic performance of laminated composites made of UHMWPE fabrics with a core layer consisting of a shear stiffening gel (SSG). SSG is a typical viscoelastic material whose mechanical properties, such as storage modulus, elastic modulus, and yield stress, are critically increased as a function of the application of external forces [193,194,195], similar to an STF mechanical behavior. The authors used boric acid (H${}_{3}$BO${}_{3}$) and hydroxyl-terminated polydimethylsiloxane (PDMS-OH) for SSG fabrication and encapsulated the fluid with 0.5 cm thickness to place between a panel, with 31 layers of UHMWPE frontal panel, 0.465 cm thick, and one layer of UHMWPE back panel. Another configuration using EVA was adopted but without further specifications. It is stated that the SSG/UHMWPE composite absorbed more energy than the neat UHMWPE panel, 120.39 and 86.46 J, respectively, indicating an increment of about 40%. Furthermore, the SSG alone absorbed 42.07 J, 109%, which is higher than the EVA alone.

Zhang et al. [148] adopted a dispersion fluid with higher molecular weight, 600 g/mol, with 50 wt% of silica nanoparticles of about 100 nm. The authors discussed the results of single- and eight-layer UHMWPE samples in both neat and STF-impregnated conditions. As expected, the STF impregnation enhanced the ballistic absorption in both samples: about 37.1% between the single-layer specimens, from 9.7 to 13.3 J and about 49.6% between the eight-layer specimens, from 127.3 to 190.5 J. Further results about energy absorption were reached, such as those relating to UHMWPE areal density and a greater number of layers, but unfortunately, no analysis was reported. Positive results were also observed in the indentation parameter for the groups analyzed, as the values reached kept under what is recommended. The eight-layer-impregnated UHMWPE presented only 13 mm of indentation, while the neat sample reached 17.5 mm.

Olszewska et al. [168] investigated the ballistic behavior of UHMWPE fabrics combined with a front layer made of elastomeric material and STF. The authors fabricated ballistic panels made of 46 layers of Dyneema, in which elastomeric anti-trauma pads (EA-TP) were inserted, as illustrated in Figure 9. The cavity of the EA-TP was filled with STF consisting of nano-size silica (particle size = 7 nm) and polypropylene glycol (PPG) with a molar mass of 400 g/mol in a 1:4 ratio. For comparison, panels with 82 layers of Dyneema were made, and both groups were evaluated. The authors subjected the samples to 7.62 × 3.9 mm caliber ammunition fire, with a velocity of 720 ± 15 m·s^−1^ and a projectile mass of 7.9 ± 0.1 g. The ballistic test results indicated that replacing 82 layers of Dyneema with 46 layers + EA-TP resulted in a 61% reduction in the penetration depth, in which the former configuration showed a BFS of 30.8 mm, compared to 12 mm for the configuration using STF.

Bajya et al. [182] focused on the ballistic performance through BFS using UHMWPE laminates and woven fabrics, as well as hybrid panels made of neat laminate layers and impregnated fabrics with SiO_2_/PEG colloidal fluid. The initial velocity of the bullet (7.5 g) was set to 430 m·s^−1^. In contrast to the above-cited papers adopting UHMWPE [18,81,148,168], the authors used the PEG fluid with a lower molecular weight of about 200 g/mol. The silica adopted for the ballistic test had a size of about 500 nm and a fixed mass fraction of 65 wt.%. The fabric was impregnated using the padding process, a technique previously mentioned [102,171], under 6 bar. All groups subjected to ballistic tests reached an areal density of about 4.5 kg/m². It is worth noticing that the authors selected the particle size as well as padding pressure through full factorial design. From the results obtained, it was observed that both the 30-layer neat UHMWPE woven fabrics and the 25-layer STF-impregnated UHMWPE woven fabrics were unable to stop the bullet from piercing. On the other hand, a hybrid set of 11 layers of unidirectional UHMWPE laminates and 12 layers of STF-impregnated UHMWPE woven fabrics was able to stop the bullets, reaching a BFS record of about 30.3 mm. The authors also highlighted that the position of the STF-impregnated UHMWPE fabrics was important for the ballistic performance, as the impregnated layers were placed at the rear side of the plate. This provides more time to trigger the shear thickening mechanism of the STF, increasing the plate strength.

Hasan-nezhad et al. [178] investigated the effect of modifying fumed SiO_2_ nanoparticles in STF using triethoxysilane on the ballistic performance of 3D glass fabrics. The fluid containing treated SiO_2_ particles dispersed in PEG 400 was denoted as treated STF (TSTF), while the triethoxysilane-free fluid was denominated as pure STF (PSTF). The 3D glass fabrics were impregnated with TSTF at 30, 40, 50, and 60 wt% concentrations and with PSTF at 30 wt%. Ballistic tests of the impregnated fabrics were performed using a 5.25 g mass and 8 mm diameter projectile, with 130 m·s^−1^ velocity. The ballistic performance of the panels containing two fabric layers was evaluated through the depth of penetration, BFS, of the projectile into the clay witness positioned behind the panel during the test. The results of the ballistic test indicated that the fabric impregnated with 30 wt% TSTF proved more effective than PSTF at the same concentration. Gradually increasing the TSTF concentration resulted in increasing improvement in ballistic performance as the BFS decreased. However, the fabric impregnated with 50 and 60 wt% TSTF showed equal values, proving that 50 wt% is enough to promote a considerable improvement in 3D glass fiber fabrics.

Unlike the studies discussed so far, few works presented potential variations of the STF, such as those presented by Liu et al. [109] and Xu et al. [24]. Both studies, based on the previous work of Wang et al. [196], Gürgen and Kushan [38], as well as Ávila et al. [100], made use of multi-phase STF for impregnation of aramid fabric and have already presented relevant results on these two works.

Liu et al. [109] explored the properties of multi-phase STF with graphene oxide (GO) and carbon nanotubes (CNTs) in SiO_2_/PEG suspensions, a combination not yet investigated. A fixed SiO_2_ content of 20 wt% was used, in which the isolated addition of 2 wt% GO, 2 wt% CNTs, and the joint addition of 1 wt% GO + 1 wt% CNTs were studied. The ballistic performance of the fabrics was evaluated by BPI, the so-called SEA in the present work, which is given by Equation (Equation 2). Ballistic tests were performed using a gas cylinder with 65 mm caliber ammunition produced in Ti${}_{6}$Al${}_{4}$V alloy with a mass of 137 g and a velocity of 250 m·s^−1^. The ballistic results indicated a superior performance of the multi-phase STF with the addition of GO, in terms of higher energy absorption, while the addition of CNT decreased the absorption ability of the impregnated fabric considerably in comparison with the single-phase STF/Kevlar. The combined addition of GO and CNTs increased the ballistic performance of the Kevlar fabric slightly, but the performance of the group is poorer than the single-phase STF/Kevlar.

In Figure 10, an interesting schematic made by Liu et al. [109] shows the deformation mechanisms of the fabric with the different types of STF, along with images of the samples after the ballistic test. Apart from the fabric impregnated with STF-GO, which remained intact after the ballistic test, all samples presented perforations in which the extension of the holes is similar to the width of the projectile. From the schematic, it is observed that the type of addition directly impacts the formation of the hydroclusters. The combination of GO and STF resulted in increased yarn friction, resulting in a greater global fabric deformation. The thickening mechanism of the STF occurred along the entire fabric, distributing the stress and reducing the damage to the Kevlar fabric. The multi-phase STF with CNT and CNT+GO displayed the thickening mechanism only in the triggered region, preventing the strain from spreading to the rest of the fabric due to lower load transfer [104,152]. As a result of the low load transfer in the weft of the fabric, the fabrics without added GO exhibited greater deformation, in addition to the presence of yarn fracture and yarn pullout.

Following the same approach, Xu et al. [24] worked on a bullet-proof vest of Twaron fabric impregnated with multi-phase STF. The fluid was composed of PEG 200, 25 wt% SiO${}_{2}$ and two types of B${}_{4}$C in different weight fractions: 5, 10, and 15 wt% for 2.45 μm B${}_{4}$C, and 10 wt% for 1.8 μm B${}_{4}$C. They were subjected to ballistic impact tests with spherical projectiles of 13 mm. Firstly, with the coarser B${}_{4}$C particles in a single-layer test, the results pointed to a superior ballistic limit of the SiO${}_{2}$/10%B${}_{4}$C/Twaron composite, which reached about 179 m·s^−1^, 11.9% higher than the SiO${}_{2}$/Twaron. The ballistic limit decreased with the addition of 15 wt% of B${}_{4}$C to about 165 m·s^−1^. Furthermore, the use of a finer B${}_{4}$C in 10 wt% decreased the ballistic limit, 162 m·s^−1^, 9.5% lower than when adopting the coarser particle in the SiO${}_{2}$/10%B${}_{4}$C/Twaron combination. All of the composites analyzed in the first step of ballistic tests presented tensile failure as the main failure mode.

In Xu et al. [24], in the multi-layer ballistic test, three layers of SiO${}_{2}$/10%B${}_{4}$C/Twaron presented the highest ballistic limit again among the impregnated fabrics, reaching 273 m·s^−1^. Although, compared to four layers of neat Twaron fabrics, the latter reached a higher ballistic limit with a 25% thinner structure. In an optimized structure, the authors compared six layers of neat Twaron with a composite made with one layer of SiO${}_{2}$/10%B${}_{4}$C/Twaron and five layers of neat Twaron. The latter presented a 10% higher ballistic limit (330 m·s^−1^) with only 5% higher areal density. Different composite sets presented a lower value of ballistic limit, with a six-layer-SiO${}_{2}$/10%B${}_{4}$C/Twaron presenting a sharp decrease, 282 m·s^−1^, which indicates that the deformation, as one of the absorbing energy mechanisms, is dramatically compromised with the STF-impregnation treatment in all layers.

A different method was also observed in the works of Haro et al. [124], Jeddi et al. [174] and Kubit et al. [179], as they made use of metallic plates to enhance the ballistic performance of the armor. The first two works [124,174] used Al plates in a sandwich set with impregnated ballistic fabric. The latter [174], in a different way, used the STF as a filler in an intermediate layer.

Haro et al. [124] used laminated Al with impregnated Kevlar, adopting STFs of SiO_2_ particles with particle size between 0.2–0.4 μm and PEG 400 as the liquid medium for particle dispersion. After fabric impregnation, the Al and Kevlar plates were bonded to the Kevlar fabrics using epoxy, as shown in Figure 11a, composing six ballistic panels configurations, on which ballistic tests were performed using 270 caliber Winchester ammunition. The authors reported that the impregnation of the fabric with STF between the layers of Al, resulted in greater integrity of the plates after the test in which, as shown in Figure 11b, a smaller deformation is observed in the samples containing STF. Another result that reinforces the positive action of the STF in this study is the results of absorbed energy, as shown in Figure 11c, which shows the superior performance of the groups of impregnated samples compared to the groups without STF treatment.

Jeddi et al. [174] used the Al sheets and fiberglass fabrics impregnated with 20 wt% SiO_2_/PEG 400 STF. Different configurations were developed to evaluate the ballistic strength of sandwich composites, which were: four layers of Al (4AL), two layers of Al + five layers of fiberglass fabric (2AL-45Neat), two layers of Al + five layers of fiberglass fabric/STF (2AL-5STF) and two layers of Al + three layers of fiberglass fabric/STF (2AL-3STF). The ballistic tests were carried out with a gas cylinder, firing 8 mm diameter projectiles with a mass of 5.25 g at a velocity of 210 m·s^−1^. The sandwich panel configuration 2AL-5STF showed better ballistic performance, followed by 5AL, 2AL-5Neat, and lastly, 2AL-3STF. The combination of two layers of Al sheets with five layers of fiberglass fabric + STF promoted a better interface, absorbing more energy during impact, about 55% higher than 2AL-3STF, and reducing the damage in the BFS. The configuration featuring only three layers of STF-impregnated fiberglass (2AL-3STF) showed the lowest ballistic result because the number of layers was not sufficient to trigger the yarn pullout energy, promoting an unsatisfactory ballistic performance. However, considering areal density, the 4AL group presented a higher specific energy absorption, about 30% higher than 2AL-5STF. This means that the 4AL presented an increase in ballistic performance without increasing much weight, and even a full metallic armor presents much lower flexibility.

As aforementioned, Kubit et al. [179] evaluated the ballistic performance of a multi-layered material made with 1.3964 stainless steel plates, aramid fabrics, and an STF-filled bag of ethylene-glycol with 5 wt% SiO${}_{2}$. The authors tested three variants: steel–fabric, steel–STF–fabric, and fabric–STF–fabric. The ballistic test was performed using 7.62 mm ammunition, with a velocity of 850 m·s^−1^. The ballistic test results showed that the addition of the STF-filled bag between the steel plate and aramid fabric layers promoted the highest drop in residual velocity after the test, although the use of a fabric–STF–fabric set presented the worst performance.

In the same way as Kubit et al. [179], but with different panel material, Cho et al. [170], in a novel application of a natural material, used the STF as a back layer of Henji papers, a paper widely known in Korea. The STF was made with cornstarch dispersed in water and was put in zipper bags for a steadier fix. The authors evaluated both the variation in the number of layers of Hanji paper and the thickness of STF layers set on the back layer, as illustrated in Figure 12a,b using a fixed number of paper layers.

The ballistic test results indicated that the addition of Hanji paper layers resulted in an increased penetrated area after firing, resulting in a low energy absorption during ballistic impact. The addition of the cornstarch/water STF considerably reduced the penetrated area as the STF layer had its thickness increased, as illustrated in Figure 12c,d.

Another research that adopted a natural material similar to Cho et al. [170] was the one by Mahesh et al. [180]. For the first time, a work explored the ballistic performance of natural fiber fabrics impregnated with STF. Unlike all the published work so far, which generally adopted synthetic fibers such as aramid, UHMWPE, E-glass, and others, STF-impregnated natural fiber fabrics had not yet been studied. The authors chose jute fabric, largely used to make carpets, packaging, textiles, rope, and other low-cost products [197]. For the composites, SiO_2_/PEG STFs were prepared in the proportions of 10, 20, 30, and 40 wt%, and poured onto panels with three and six layers of jute fabric. The ballistic performance was compared to neat jute fabrics also with three and six layers. The ballistic tests were performed using 9 mm ammunition, according to the IIIA level of NIJ standard 0108-01 [198]. It is interesting to note that the STF-impregnation indeed reinforced the natural fabric under ballistic impact. As expected, the results of the ballistic tests indicated that six-layer samples showed a superior ballistic limit and superior energy absorption for all sample groups compared to three layers. A significant enhancement was observed in up to 30 wt%-impregnated fabric, with a negligible increase between 30 and 40 wt% nanosilica loading, 3.8 and 4.1% for three and six layers, respectively. Another interesting parameter is that the specific energy absorption of the 30 wt%-impregnated composites is highly superior to neat jute fabric and similar to the 40%-impregnated composite. From the morphology of the damaged fabrics, also illustrated in Figure 13, it is observed that the addition of STF at proportions of 10 and 20 wt% promoted mechanical interlocking and reduced fiber pullout. Additions of 30 and 40 wt% did not show fiber rupture and pull out, besides preventing the penetration of the projectile.

The use of natural fibers for ballistic application has been extensively studied in different sets [199,200,201], with a huge potential for acceptable performance. Further remarks are discussed in the following section.

## 4. Future Trends

The current state-of-the-art of natural woven fabric points to substantial growth in ballistic analysis, especially those made with lignocellulosic fibers [200,202,203,204]. In fact, natural vegetable fibers are renewable materials composed of cellulose chains, as well as hemicellulose and lignin. Thus, this kind of material presents some interesting remarks, such as low density, high specific strength, huge availability, and variety [205,206,207]. For soft armor applications using the STF-impregnation technique, only Mahesh et al. [180] have explored this combination so far. As aforementioned, an impressive approximate 11-fold increase in energy absorption and a 3-fold increase in ballistic limit was observed between the six-layer neat jute and six-layer impregnated-jute-fabric with 40 wt% of SiO${}_{2}$, although further improvement is needed to reach the ballistic performance of a full-synthetic armor vest. Therefore, a deeper investigation into the use of natural lignocellulosic fabrics impregnated with STF may push the boundaries of soft armors with cheaper, greener, and equally strong solutions.

A similar approach to natural alternatives is the mineral basalt fiber fabrics, which were explored recently under low-velocity impact [208,209]. Sun et al. [209] explored the use of STF-impregnation with 20 wt% of silica on the basalt fabric and found an improvement of about 12% of the absorbed energy. Kumar et al. [208] used silica/epoxy impregnation to increase the yarn friction of an aramid/basalt hybrid armor and reached an increase in impact strength of about 15%. Both works revealed the valuable potential of the fabric under impact and must be an alternative for further studies.

Another novelty was spotted in the work of Gürgen and Kushan [210] with the use of STF-impregnated ballistic fabric for the improvement of spall liner performance. According to the authors, spall liners, mainly used in armored vehicles, are responsible for restraining the particles of projectiles after impact with a frontal armor layer, subject to a multi-point ballistic impact. Thus far, glass, Kevlar, and hybrid carbon-Kevlar fabric [211], as well as hybrid kenaf-aramid fabric [212], were investigated, aiming to obtain a potential weight, thickness, and material reduction.

Recently, two works about STF-impregnation auxetic fabrics also presented remarkable performance under low-velocity impact tests [213,214]. Auxetic materials are those with a negative Poisson ratio. In fact, auxetic material contracts in the transverse direction when compressed and expands to all directions when pulled in only one [215]. In this first work, the authors adopted auxetic fabrics of polyester multifilament to form rotating hexagonal meshes and STF of fumed SiO${}_{2}$ and PEG200, 400, and 600. The use of the STF-impregnation method has interestingly enhanced the energy absorption when compared with neat fabric, associated with the increase in weight fraction of silica nanoparticles. The same occurred with Pais et al. [214], adopting polyamide and elastane as raw materials for the fabric. Thus, according to the results reached and the low number of publications, the STF-impregnated auxetic fabric presents promising potential for future works and ballistic applications.

## 5. Final Remarks

This review article provides an updated overview of impregnated ballistic fabrics for soft armor vest application. As aforementioned, since the emergence of STF-impregnated fabrics in the early 2000s, several works have started to be developed in order to improve the ballistic performance along with weight decrease. Thus, these features were analyzed based on materials adopted, methodologies applied, and results achieved, along with a brief theoretical background to base concepts about the thickening mechanism as well as particle and fabric characteristics.

The present work, in general terms, contributed to the knowledge organization through an overview of the unexplored state-of-the-art, which is essential for a full understanding of the composite behavior. In specific terms, it could be noted from the works that silica particles mixed in PEG 200 were largely adopted in Aramid fabrics and presented acceptable results. To a lesser extent, different materials, such as B${}_{4}$C and GO, were applied to improve the shear thickening behavior provided by the SiO${}_{2}$/PEG fluid and achieved even better performance. The same improvement was found in those works that selected UHMWPE, which was scarcely analyzed as part of an STF-impregnated ballistic panel system.

From this scenario, a desirable tendency of future scientific investigations is the adoption of different material combinations, as some recent works reported have already been conducted with B${}_{4}$C, GO, CNT, cornstarch, etc. In addition, a homogenization of parameters related to ballistic tests, such as initial velocity and projectile mass, would enrich a future comparative ballistic analysis of different materials. Furthermore, a deeper understanding of the fluid dynamics through the ballistic panels as a function of time and temperature is relevant for the durability analysis of the soft armor vest, as well as a simpler impregnation process.

Nonetheless, the results obtained so far ratify a potential adoption of STF for ballistic strength enhancement of fabrics, aligned to a simple and cost-effective solution for the production of soft armor. Therefore, the state-of-the-art reported so far could demonstrate that the ballistic application based on NIJ standards is undoubtedly feasible.

## Figures and Tables

**Figure 1 polymers-14-04357-f001:**
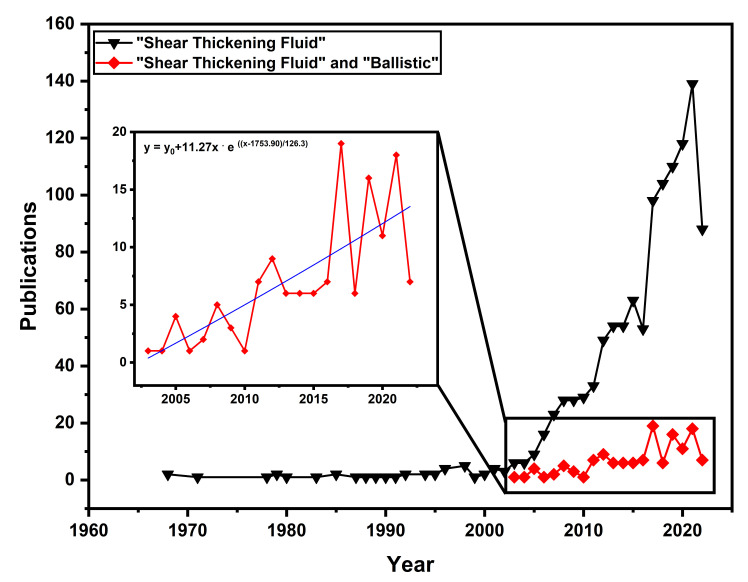
Publications per year for different keywords using the Scopus database. The black line represents the number of publications with the keyword “Shear Thickening Fluid”, while the red line represents the keywords “Shear Thickening Fluid” and “Ballistic”.

**Figure 2 polymers-14-04357-f002:**
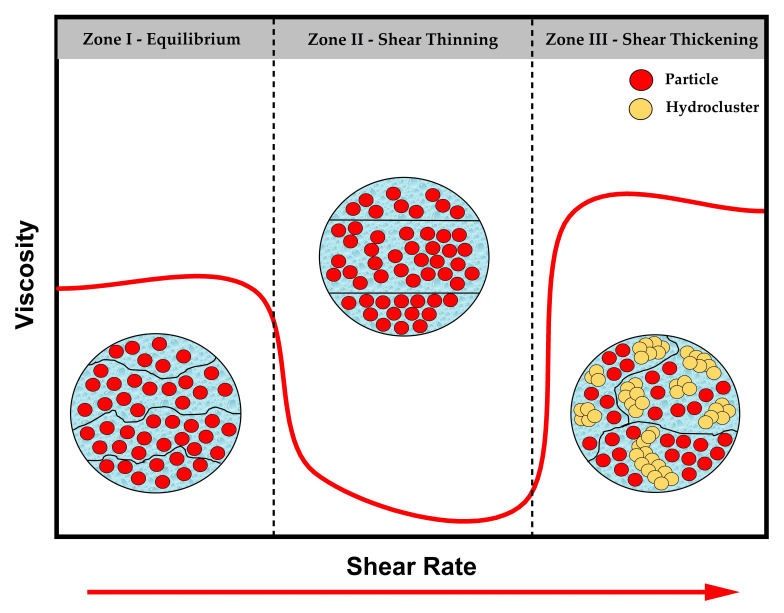
Schematic representation of the rheological behavior of a typical STF.

**Figure 3 polymers-14-04357-f003:**
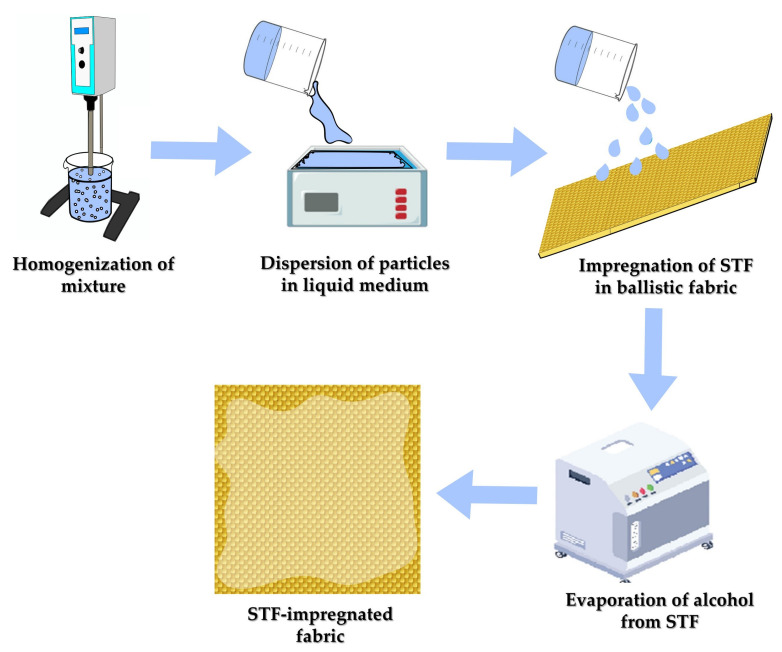
Scheme of preparation method of shear thickening fluids.

**Figure 4 polymers-14-04357-f004:**
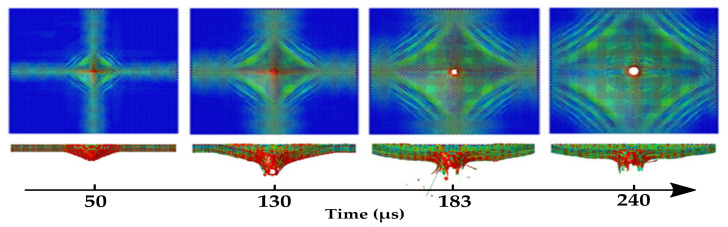
Visual aspect of the cross strain and pyramidal elevation due to stressed yarns and inter-yarn friction of a Kevlar fabric during ballistic impact. Adapted with permission from Ref. [105]. Copyright 2015, Elsevier.

**Figure 5 polymers-14-04357-f005:**
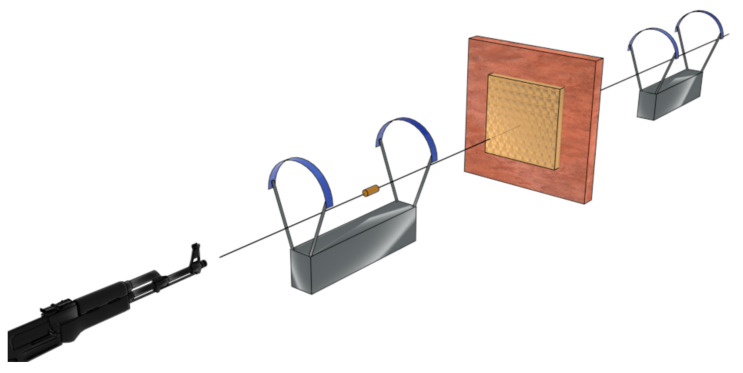
Schematic figure of ballistic test apparatus.

**Figure 6 polymers-14-04357-f006:**
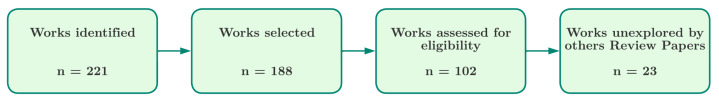
Diagrams of the selection process and the number of works found based on previous review papers.

**Figure 7 polymers-14-04357-f007:**
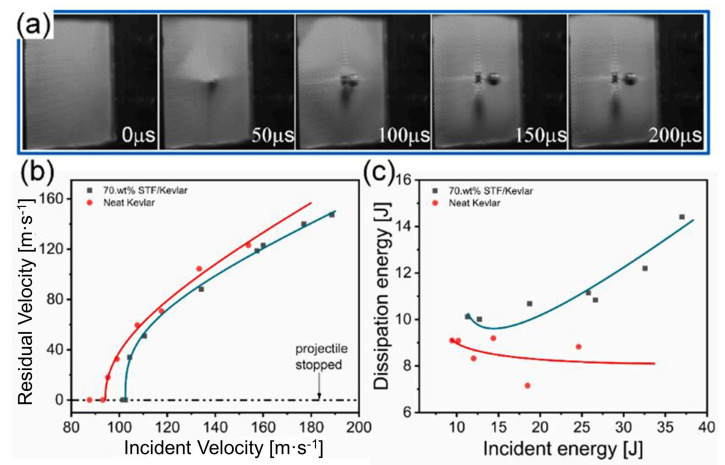
(**a**) The impact process diagram of neat Kevlar with different incident velocity; (**b**) the residual velocity results of STF/Kevlar and neat Kevlar; (**c**) the corresponding energy dissipation results. Reprinted with permission from Ref. [176]. Copyright 2021, Elsevier.

**Figure 8 polymers-14-04357-f008:**
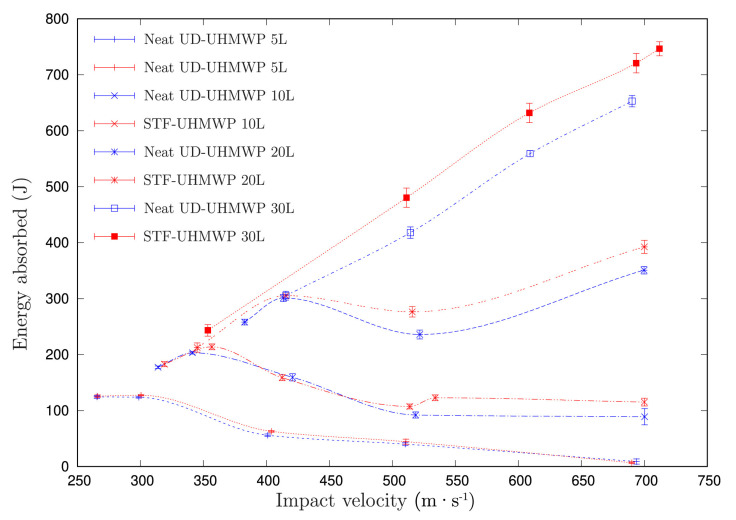
Variation of energy absorbed with the impact velocity of completely penetrated neat and STF-impregnated panels of different layers. Reprinted with permission from Ref. [81]. Copyright 2022, Elsevier.

**Figure 9 polymers-14-04357-f009:**
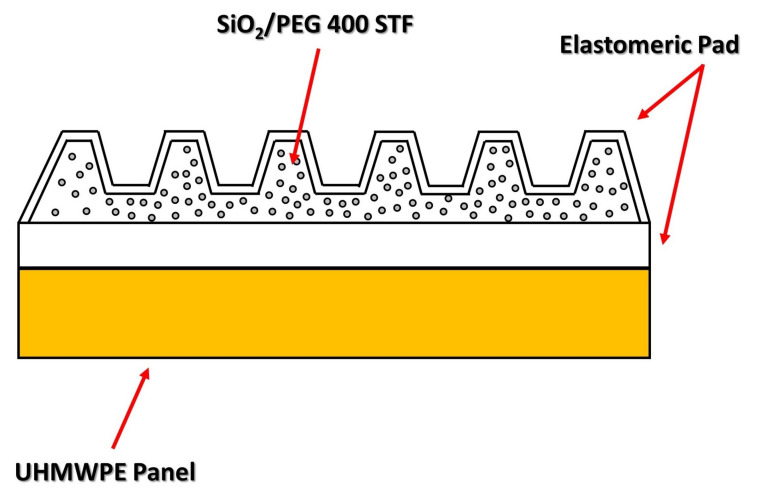
Schematic of the ballistic plate composed of Dyneema panels + elastomeric anti-trauma pads with shear thickening fluid. Adapted with permission from Ref. [168]. Copyright 2017, Sciendo.

**Figure 10 polymers-14-04357-f010:**
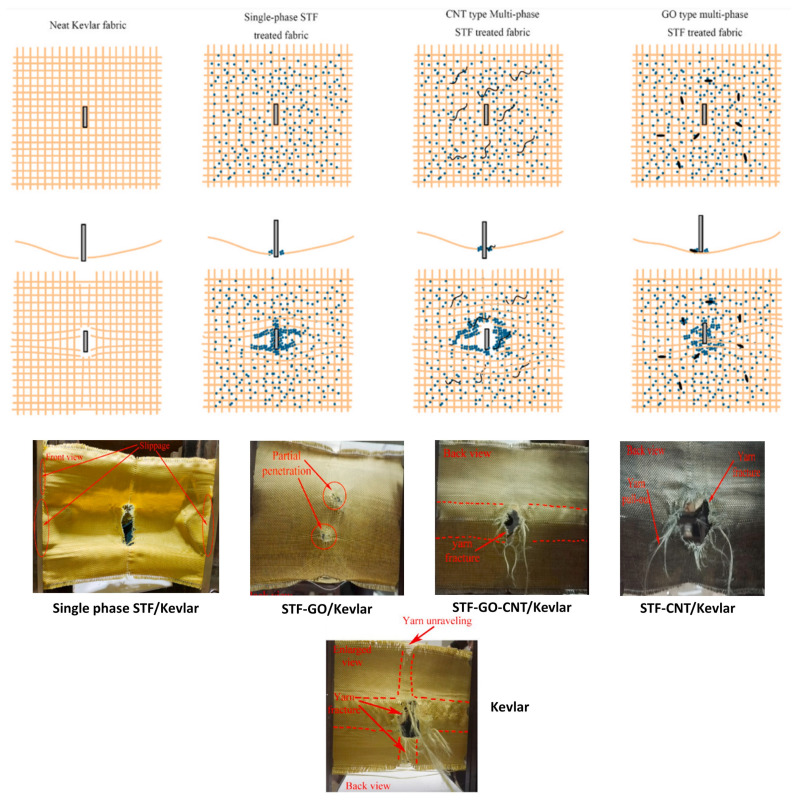
Schematic of the improvement mechanism of multi-phase STF treated fabric and damage visualization of single-phase and multi-phase STF-impregnated Kevlar fabrics after the ballistic test. Adapted with permission from Ref. [109]. Copyright 2022 Elsevier.

**Figure 11 polymers-14-04357-f011:**
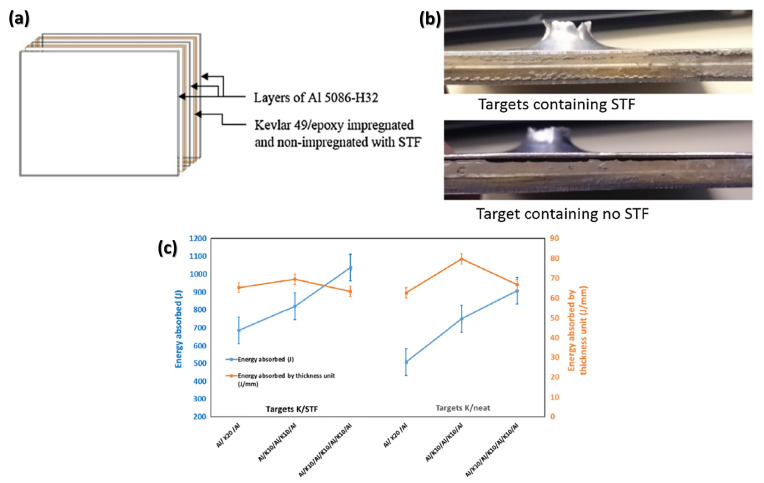
(**a**) Schematic of the arrangement of the Kevlar and Al layers; (**b**) side view of the samples after the ballistic test indicating the integrity of the samples; (**c**) relation between energy absorption by thickness unit and dissipated energy after ballistic impacts at the specimens. Adapted with permission from Ref. [124]. Copyright 2016, Elsevier.

**Figure 12 polymers-14-04357-f012:**
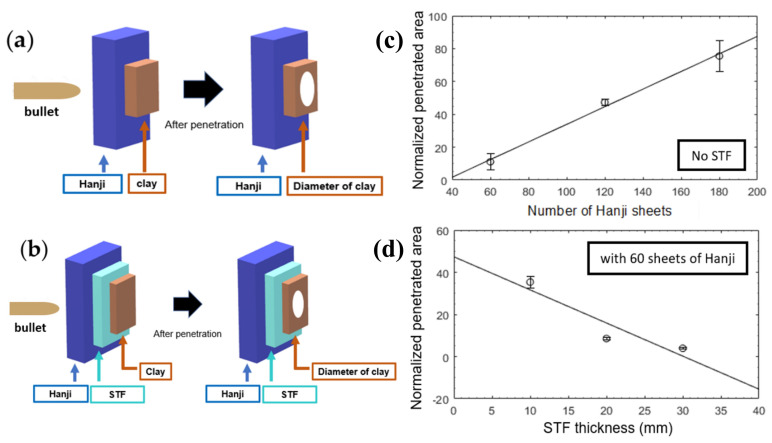
Schematics illustrating the position of the panels used in the ballistic test: (**a**) composites with only Hanji paper layers; (**b**) hybrid composites composed of Hanji and cornstarch STF; (**c**) graph of the normalized penetrated area in the panels as a function of the number of Hanji paper layers; (**d**) graph of the normalized penetrated area in the panels as a function of the STF thickness. Adapted with permission from Ref. [170]. Copyright 2020, MDPI AG.

**Figure 13 polymers-14-04357-f013:**
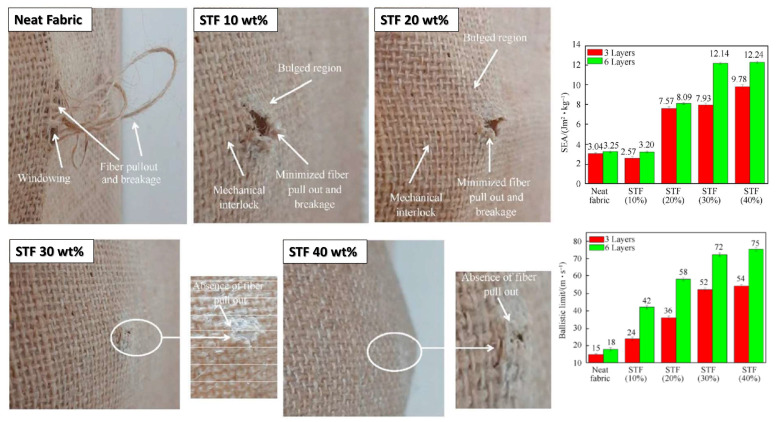
Damaged morphologies of neat jute fabric and jute fabric impregnated with different concentrations of STF. Plots of specific energy absorption (SEA) and ballistic limit of neat jute fabric and impregnated fabrics. Reprinted with permission from Ref. [180]. Copyright 2022, Elsevier.

**Table 1 polymers-14-04357-t001:** Test requirements from NIJ 0101.06 standard [13].

Armor Type	Test Bullets	Bullet Mass (g)	Armour Test Velocity (m·s^−1^)
IIA	9 mm FMJ RN 0.40 S & W FMJ RN	8.0 11.7	373 352
II	9 mm FMJ RN 0.357 Magnum JSP	8.0 10.0	398 436
IIIA	0.357 SIG FMJ FN 0.44 Magnum SJHP	8.1 15.6	448 436
III	7.62 mm NATO FMJ	9.6	847
IV	0.30 Calibre M2 AP	10.8	878

**Table 2 polymers-14-04357-t002:** Summary of theories to explain shear thickening fluids.

Theory	Explanation	Reference
Order–Disorder	The repulsive forces between particles keep them separated and ordered in layers. The external force can exceed this interaction and cause a disorder in the structure, which increases the viscosity of the material	[49]
Hydrocluster Formation	The thickening occurs due to the formation of clusters from the fluid force, which complicates the flow of the dispersion liquid and increases the viscosity of the material	[50]
Contact Rheology Model	Takes into account both the lubrication forces and solid friction among particles, the latter considered a dominating force. The result of this model allows predicting the abrupt increase in the shear thickening behavior.	[2]

**Table 3 polymers-14-04357-t003:** Summary of factors influencing the rheological properties of shear thickening fluids.

Phase	Factor	Description	Reference
Particle	Fraction	The increase in particle concentration (in wt. %) increases the viscosity of the STF. In the case of nanoparticles, the increase in concentration causes an increase in hydrodynamic forces due to the reduction in the distance between the particles, increasing the number of hydroclusters at small shear rates.	[39,49,50,66,67]
	Aspect ratio	Particles with high aspect ratios are more propitious to increase particle interlocking and rotational movements in the flow. They have a greater possibility of contact with nearby particles, triggering STF thickening more effectively.	[47,68,69,70]
	Size	Smaller particles increase the viscosity of mixtures due to the increased number of particles per unit volume in finer particle dispersions, resulting in a higher interparticle bond density. The study of Brownian forces should be considered in the investigation of this trend since they dominate nano-sized suspensions and retard thickening for higher shear rates because of enhanced repulsive stresses between the particles.	[46,71,72,73]
	Particle–particle interactions	During shear thickening behavior, the particles may remain neutral and repel each other due to entropic or stereoscopic interactions. Deflocculated suspensions have low viscosity at low thickening rates. The flocculated suspensions, on the other hand, have high viscosity at low shear rates.	[47,70]
	Hardness	Harder particles are suggested for the development of STFs as they lead to a better shear thickening mechanism due to their enhanced mechanical properties. In the interaction of the particles, the ones with low hardness cannot support the increased stresses, resulting in a drop in viscosity.	[74,75]
Liquid Medium	Molecular Weight	Higher molecular weight fluids used in STFs exhibit higher viscosity in lower shear rates due to longer molecular chains, which makes it difficult for adjacent layers of fluid to move closer to each other. Thickening is accomplished by increased adsorption of polymers on the surface of the particles due to polar interactions between silanol groups present on the silica particles with long polymer chains, increasing adsorption.	[76,77,78]
	Temperature	The viscosity of suspensions decreases with increasing temperature. The reduction occurs due to reduced hydrogen bond strength. In addition, Brownian motion is enhanced, disordering the thick structure and delaying the critical shear rate to higher values.	[50,66]

**Table 4 polymers-14-04357-t004:** Summary of factors influencing the ballistic performance of ballistic fabrics.

Factors	Description	Reference
Fiber type	Low density, high tensile strength, and modulus are important criteria for high-velocity impact strength and soft armor application. They are related to high crystalline orientation, long molecular chain length, and strong chemical bonds, which lead to improved performance. Tabiei states that the higher the modulus and the lower the density, the faster the stress will propagate through the fabric, leading to an increase in energy dissipation.	[2,88]
Areal Density	A fabric with a low number of yarns per unit dimension may present a lower impact energy absorption due to yarn sliding, causing the “wedge through”. On the other hand, high areal density affects the final weight and may raise yarn stress concentration, which induces a fiber break. A moderate grammage (between 0.65 and 0.95) maximizes the area activated under the impact, which increases the ballistic energy absorption.	[89]
Weave Patterns and Crimp	Weave patterns used in ballistic applications are usually plain and basket weaves. Although, plain-woven fabrics present higher undulation, resulting in a superior abrasion and energy dissipation, called yarn crimps.	[88]
Number and Orientation of Layers	Energy absorption increase with multiple layers. Friction between the plies inhibits the motion of the yarns, resulting in a higher impact strength. Furthermore, the effect of bullet geometry decreases as the number of plies increases.	[90,91]

**Table 5 polymers-14-04357-t005:** Review works screened and read for report selection.

Author	Year	Reference	Cited Works
Thilagavathi et al.	2008	[110]	[35,69,111]
Srivastava et al.	2012	[70]	[35,69,70,72,74,94,112]
Ding et al.	2013	[113]	[35,114]
Hasanzadeh et al.	2014	[115]	[35,69,72,116,117]
Bilisik	2017	[79]	[35,70,94,118]
Gürgen et al.	2017	[46]	[35,41,69,72,82,115,117,119,120,121,122,123,124]
Mawkhlieng and Majumdar	2019	[2]	[34,35,46,69,74,82,94,104,116,117,124,125,126,127,128,129,130,130,104]
Mawkhlieng et al.	2020	[15]	[23,35,69,70,72,104,115,117,125,131]
Weerasinghe et al.	2020	[108]	[35,38,39,41,69,72,84,100,119,120,121,132,133]
Zarei and Aalaie	2020	[45]	[35,38,40,41,69,72,83,119,123,133,134]
Abtew et al.	2021	[135]	[35,37,38,39,41,72,115,117,121,123,136,117,38,121]
Czech et al.	2021	[80]	[79,100,123,134,136,137,138]
Khodadadi et al.	2021	[42]	[26,35,38,39,41,79,83,84,88,100,119,121,122,123,137,138,139]
Muneer-Ahmed et al.	2021	[140]	[35,37,69,84,108,121,141,142,143,144,144]
Zhang et al.	2021	[43]	[37,38,39,40,41,82,83,89,100,109,116,123,124,129,145,146,147,148,149,150,151,152]
Wei et al.	2022	[153]	[36,38,39,72,94,100,104,116,117,121,125,133,134,137,149,152,154,155,156]
Zhang et al.	2022	[44]	[23,26,35,41,44,46,84,100,108,117,119,125,129,132,147,149,157,158,159,160,161,162,163,164,165]

**Table 6 polymers-14-04357-t006:** Summary of materials used in unexplored publications related to shear thickening fluid and ballistic fabrics.

Authors	Panel	Particle	Liquid Medium	Content
Levinsky et al. [166]	Aramid	Al_2_O_3_	PVAc	3 wt%
Grineviciute et al. [167]	Aramid	H_4_SiO_4_ Acrylic	Water	5 wt%
Haro et al. [124]	Aramid + Al plates	SiO_2_	PEG 400	50 wt%
Olszewska et al. [168]	UHMWPE	SiO_2_	PPG 400	25 wt%
Tria et al. [169]	Aramid	SiO_2_	PEG	-
Cho et al. [170]	Hanji Paper	Cornstarch	Water	55 wt%
Yeh et al. [171]	Aramid	SiO_2_	PEG 200	20, 30, 40, and 50 wt%
Yeh et al. [172]	Aramid	SiO_2_	PEG 200	40 wt%
Liu et al. [109]	Aramid	SiO_2_ GO CNT	PEG 200	20 wt%
Chang et al. [173]	Aramid	SiO_2_	PEG 200	40 wt%
Jeddi et al. [174]	E-glass + Al plate	SiO_2_	PEG 400	20 wt%
Katiyar et al. [175]	Aramid	SiO_2_	PEG 400	25 wt%
Zhang et al. [176]	Aramid	SiO_2_	PEG 200	0, and 70 wt%
Bablu and Naminala [177]	Aramid	SiO_2_	Water	0, 10, 30, and 40 wt%
Hasan-Nezhad et al. [178]	E-Glass	SiO_2_	PEG 400	30, 40, 50, and 60 wt%
Kubit et al. [179]	Aramid + Steel plate	SiO_2_	EG	5 wt%
Mahesh et al. [180]	Jute Fabric	SiO_2_	PEG	10, 20, 30, and 40 wt%
Mishra et al. [81]	UHMWPE	SiO_2_	PEG 400	40 wt%
Tang [18]	UHMWPE	-	H_3_BO_3_ + PDMS-OH (SSG)	-
Xu et al. [24]	Aramid	SiO_2_ B_4_C	PEG 200	SiO_2_ – 0, and 25 wt% B_4_C – 0, 5, 10, and 15 wt%
Zhihao et al. [181]	Aramid	SiO_2_	PEG 200	20 wt%
Zhang et al. [148]	UHMWPE	SiO_2_	PEG 600	50 wt%
Bajya et al. [182]	UHMWPE	SiO_2_	PEG 200	65 wt%

## Data Availability

Not applicable.

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
