# Peer review of "Fabric Impregnation with Shear Thickening Fluid for Ballistic Armor Polymer Composites: An Updated Overview"

_polymers, 2022, doi:10.3390/polym14204357_

Round 1

Reviewer 1 Report

This is a very unique manuscript, providing a clear review on the works that haven't been reviewed so far in the field of shear thickening fluid applied in ballistic armor polymer composites. The scope of the review is well defined, and the literature searching methodology is clearly presented. The authors give an exhaustive introduction on the recent progress and outlook in this field. I would suggest the acceptance of this work after minor correction of some grammatical errors.

Line 65, excessive number of layer(s) 

Line 170, first two mechanisms

Line 186, the ballistic tests cover well..

Line 360, Their ballistic tests have been...

Line 373, Polyvinyl Acetate (PVA), I understand that polyvinyl acetate can be abbreviated as PVA, but I would recommend that the authors can correct it as PVAc. The reason is that PVA is more commonly referred to as polyvinyl alcohol. PVAc is less misleading.

Out of curiosity, in Table 6, citation 168, what is acrylic water particle?

Overall, this is a very nice written manuscript.

Author Response

Dear Reviewer,

Sincerely,

Matheus Pereira Ribeiro

Reviewer 2 Report

Revise the paper according to the attached file.

Author Response

(The authors gave the same response as above.)

Reviewer 3 Report

Manuscript ID: polymers-1969299

Review Report

The reviewer would like to submit the review report to the manuscript entitled " Fabric Impregnation with Shear Thickening Fluid for Ballistic Armor Polymer Composites: An Updated Overview," which might be considered for publication in the Polymers (ISSN 2073-4360).

Following the completion of the review process, the reviewer would like to provide some critical thoughts and ideas to assist the authors in working more effectively.

Comments and Suggestions for Authors

In this study, the authors have an outlook of layered, high-strength woven fibers that make up soft armor to protect against medium-caliber missiles without compromising mobility. In wars, many fabric layers are needed, increasing armor weight, thickness, and stiffness and limiting fighter mobility. In the early 2000s, armor fabric panels were coated with non-Newtonian colloidal shear thickening fluid (STF). STF-impregnated fabrics give flexible ballistics. These reviews need to be updated to include more literature. Updated summaries must fix gaps and cover missing articles. This overview addresses STF-impregnated fabrics as easy, cost-effective approaches for generating soft armors with improved ballistic fabric performance. The summary states that NIJ-based ballistic applications for these materials are possible. The reviewer concluded that the paper has potential and could be published. Please revise based on the reviewer's comments.

General comments

The article is generally arranged scientifically, logically, and coherently. The technical English language used is relatively straightforward to understand. Many tables and pictures summarize the research so far in scientific order, beautifully arranged, and easy to understand. Besides that, there are still some points to note below.

ü  Keywords: there are two keywords( Shear Thickening Fluid and STF) that are the same. Please delete the one.

ü  Figure 1: Please correct the equation in the figure. (multiply and division signs are wrong)

ü  Table 1: Please add the dot (.) between the units of m and s-1

ü  The authors remark: “The research collection was updated until August 2022 through the electronic databases: SCOPUS, Web of Science, Google Scholar, SciELO, SAGE Journals Online, ScienceDirect.” The reviewer suggests that (with the trust database) SAGE Journals Online and ScienceDirect belong to SCOPUS or Web of Science (database); therefore, please remove SAGE Journals Online and ScienceDirect from this list. The journals/ documents out of the SCOPUS or Web of Science (database) should consider reliable sources or not.

ü  Figure 4: Please add the time on the horizontal axis for each specific period shown in the figure. (50, 130, 183 and 240 µs)

ü  Equation (1): please clarify the symbol v (lowercase) as considering standing for velocity and V(uppercase) that considering standing for volume.

ü  Figure 7: Please consider the incident velocity unit in the figure that m/s should be m.s-1

ü  Lines 258, 295, 306, 316, 327, 329, 334, 345, 397, 404, 406, 408,443, 451, 472, 492, 517, 518, 519, 524, 527, 529, 554, 569: m/s should be m.s-1

ü  Figure 12: there is very poor image quality. Please increase the resolution of the image to make it easier to read.

2. Questions:

2.1 As shown in Figure 6, please explain the sources of unexplored reports (n=20). Are these sources mostly national military secrets? How can you access them?

2.2 Please add research gaps (not in this study, but for future research) in the conclusions to guide further research. And the decisions and comments of the authors' findings profoundly influence the development of this research field in the present and future.

2.3 Regarding the reused figures in this study, have the authors asked permission from the authors of the original figures? If yes, please provide proof in the supplementary document.

The reviewer enjoys reading the manuscript.

Thank you.

Sincerely yours,

The Reviewer

Author Response

(The authors gave the same response as above.)

Round 2

Reviewer 2 Report

I think the authors uploaded the old version of the manuscript.

Please, upload the revised paper and highlight the added parts.

Reviewer 3 Report

The authors have answered all my question. I suggest the manuscript should be accepted for publication 

Thank you.

Round 3

Reviewer 2 Report

For the second time, the authors upload a wrong file.

The manuscript has not been updated (no color highlighting). 

Round 4

Reviewer 2 Report

For the second time, the authors upload a wrong file.

The manuscript has not been updated (no color highlighting). 

I think the authors uploaded an old draft of the manuscript.
